# Correlation between Antimicrobial Activity Values and Total Phenolic Content/Antioxidant Activity in *Rubus idaeus* L.

**DOI:** 10.3390/plants13040504

**Published:** 2024-02-11

**Authors:** Audrone Ispiryan, Vilma Atkociuniene, Natalija Makstutiene, Antanas Sarkinas, Alvija Salaseviciene, Dalia Urbonaviciene, Jonas Viskelis, Rasa Pakeltiene, Lina Raudone

**Affiliations:** 1Agriculture Academy, Vytautas Magnus University, Studentu Str. 11, LT- 53361 Akademija, Lithuania; vilma.atkociuniene@vdu.lt (V.A.); rasa.pakeltiene@vdu.lt (R.P.); 2Food Institute, Kaunas University of Technology, Radvilėnu av. 19 C, LT-50254 Kaunas, Lithuania; natalija.makstutiene@ktu.lt (N.M.); antanas.sarkinas@ktu.lt (A.S.); alvija.salaseviciene@ktu.lt (A.S.); 3Institute of Horticulture, Lithuanian Research Centre for Agriculture and Forestry, Kaunas Str. 30, LT-54333 Babtai, Lithuania; dalia.urbonaviciene@lammc.lt (D.U.); jonas.viskelis@lammc.lt (J.V.); 4Laboratory of Biopharmaceutical Research, Institute of Pharmaceutical Technologies, Lithuanian University of Health Sciences, Sukileliu Av. 13, LT-50162 Kaunas, Lithuania; lina.raudone@lsmu.lt

**Keywords:** plant by-products, characteristics of raspberry morphological parts, antimicrobial activity, phenolic compounds, added value, antibacterial products

## Abstract

Plant by-products, which are discarded into the environment, are rich in valuable compounds. The aim of this research was to determine the antibacterial activity of *Rubus idaeus* L. morphological parts and its correlation with total phenolic content and antioxidant activity. The authors also aimed to evaluate the plant’s potential as added-value products. New aspects were revealed for further use and for making novel and natural products. The study’s results indicated that raspberry leaves, inflorescences, and fruits could effectively combat three Gram-positive bacteria. According to the findings, among the various plant parts, root and seed extracts had the lowest antibacterial activity. Data revealed moderate, weak, or very weak correlation between the antimicrobial activity and phenolic content parameters. These findings underscore the viability of substituting synthetic antimicrobials with natural alternatives. The present study is significant for preparing novel products as antibacterials by appropriate and optimized processing using all raspberry morphological parts, and the research results show promising prospects for future purposeful utilisation of nature-based products. Raspberry plant parts can find applications in emerging fields that generate economic and environmental value.

## 1. Introduction

In the pursuit of mitigating the impact of harmful bacteria, there is a global exploration of natural herbs with historical ethnomedicinal use. Investigations aim to establish their pharmacological effects based on traditional knowledge and practices [1,2]. Bacterial infections are among the prominent causes of health problems, physical disabilities, and mortality around the world. In recent decades, the proliferation of bacteria resistant to multiple antibiotics has become increasingly widespread [3]. This trend has posed challenges to effectively managing and treating certain human infectious diseases.

Since ancient times, people have treated themselves with natural remedies, and many modern medicines are based on the knowledge of traditional medicine. Plants are often used for the treatment of diseases due to their valuable effects on health. Therapeutic use makes them popular and acceptable across all culture and religions for implementation in healthcare all over the world [4]. In recent years, there has been a growing interest in plant phytochemicals as alternative strategies for the treatment of bacterial infections. The use of natural medicines from plant raw materials as an alternative to antibiotics is attractive because bacterial resistance to a mixture of active molecules can develop more slowly than that to the action of a single compound, as is usually used in antibiotic treatment [5]. Also, plant extracts preserve probiotic species in the microbiota [6,7]. It is a safe and cost-effective way of treating bacterial infections.

The World Health Organization (WHO) [8] provides a global priority pathogens list (PPL), recommending that research and development should be directed towards uncovering effective antibiotic treatment pathways. Scientists have conducted many studies that prove that natural antimicrobial compounds from plants can be used as an alternative preservative and pathogen control method against bacteria [9]. However, there is still a lack of knowledge about the antioxidant activity and properties of many plants that would help in the fight against diseases caused by bacteria. 

Natural antimicrobials are secondary metabolites that can be produced by living organisms, including plants, animals, and microorganisms. The antimicrobial activity of these metabolites has only been scientifically confirmed in the last 30 years [10]. There are four major groups of antimicrobial compounds made by plants: phenolics and polyphenols, terpenoids and essential oils, lectins and polypeptides, and alkaloids. The rising demand for natural antimicrobials in the food and beverage industry driven by their advantages, such as high efficacy, eco-friendliness, and lack of side effects, is one of the major market drivers credited with the market’s expansion. Demand for natural antimicrobials is also being fuelled by the increasing awareness of the health benefits associated with consuming foods with minimal to no preservatives [11]. The market for natural antimicrobials has greater growth potential as a result of rising health awareness among consumers of organic food and clean-label products. 

Raspberries (*Rubus idaeus* L.) are members of the *Rosaceae* family. Numerous research studies have outlined that raspberries have beneficial properties for humans, like antioxidant and antimicrobial activities, and a wide range of physiological properties, such as anti-allergenic, anti-atherogenic, anti-inflammatory, antimicrobial, antioxidant, and cardioprotective effects. The pharmacological activity of natural compounds is due to their low toxicity, the ability to comprehensively affect the body, and rarely causing serious adverse reactions [12,13,14,15,16,17,18,19]. The antimicrobial potential of some raspberry morphological parts, such as fruits and leaves, has been investigated mostly by the scientists mentioned in this paper. Some scientific works on raspberry inflorescences and seeds can also be found. However, there is still a lack of research regarding raspberry stems and roots, and there are no scientific data comparing all raspberry morphological parts cultivated under the same conditions. Understanding the biological activity mechanisms of raspberry and its constituents is crucial to establishing it as a potential new source of antibiotics. However, based on our literature research, there are not enough data available regarding the antimicrobial activity of all raspberry plant morphological parts cultivated under the same conditions [20,21,22,23].

The aim of this study was to determine the antimicrobial properties of raspberry fruit, leaf, stem, inflorescence, seed, and root extracts against Gram-positive (*Staphylococcus aureus*, *Bacillus subtilis*, *Enterococcus faecalis*, and *Listeria monocytogones*) and Gram-negative (*Salmonela Typhimurium*, *Pseudomonas aeruginosa*, *Enterobacter aerogenes*, and *Escherichia coli*) bacteria. Additionally, we aimed to establish an correlation between antibacterial activity and total phenolic content/antioxidant activity, exploring potential applications for added-value products.

## 2. Results and Discussion

### 2.1. Bacterial Feature Analysis

Our research showed that ethanolic extracts from the fruits, roots, stems, seeds, leaves, unripe fruits, and inflorescences of raspberry “*Polka*” are effective against *Staphylococcus aureus*, *Listeria monocytogenes*, *Salmonella typhimurium*, *Bacilus subtilis*, *Enterococcus faecalis*, and *Pseudomonas aeruginosa*.

For comparison, Ryan et al. investigated the antimicrobial properties of raspberry leaf, juice, and commercial leaf tea against five human pathogenic bacteria and two fungi. Their research results revealed that raspberry cordial and juice reduced the growth *of Salmonella, Shigella*, and *E. coli* but did not show antifungal activity. Contrary to the results of our study, they reported that no antimicrobial activity was detected in the leaf extract or tea [24]. However, our study refutes this, as the leaves had very high antimicrobial activity against *L. monocytogenes* and *S. aureus* bacteria.

To assess the antimicrobial testing of unripe raspberries, numerous studies have focused on raspberries grown in China, specifically the unripe (green) raspberry of *Rubus chingii Hu*. It is worth mentioning that in a recent study, Jiang et al. (2021) [25] extensively studied the antimicrobial potential of these unripe raspberries, including unpurified raspberry extract and purified raspberry extract (*Rubus chingii Hu*), against *Escherichia coli, Staphylococcus aureus*, and *Salmonella enterica*. Dutreix et al. (2018) [26], in their work, highlighted for the first time the potential of ripe and unripe fruits of *R.idaeus* to prevent oral *C.albicans* biofilms. Their study focused on red raspberry fruit, known for its richness in tannins with potential antimicrobial properties. Ördögh et al. [27] tested the antibacterial activity of raspberry juice. The results showed antibacterial activity only against *Staphylococcus epidermidis*. The juice did not affect the growth of *S. aureus, Streptococcus pyogenes*, and *Propionibacterium acnes*, while pomace extracts did not show any antibacterial activity. Similarly, Bobinaite et al. [28] determined the antimicrobial properties of raspberry fruit, pulp, and marc extracts from different plant cultivars. They declared the same findings as the results reported by Puupponen-Pimia et al. [29,30], who concluded that berry phenolics were only partially responsible for the growth inhibition of *Salmonella* and that most of the antimicrobial effects originated from other compounds, such as organic acids.

Leaves are less studied than fruits probably because of the wide use of fruit in human nutrition. However, the antimicrobial and antiviral properties of plants have also begun to be more actively researched after the breakthrough in packaging production in recent years, as evidenced by various research studies on films [31,32]. Kucharskii et al. [33] stated that the raspberry leaves of the genus *Rubus* can be applied as antiseptic agents in the treatment of skin diseases. The biological action exerted by the R.idaeus leaf hydrolate, investigated by De Santis et al., was observed on *B. cereus* and *A. bohemicus* strains [34]. Veljković et al. tested the antimicrobial activity of the methanolic extracts of the leaves and fruits of wild raspberry against Gram-positive *Sarcina lutea* and *Bacillus subtilis* and Gram-negative *Escherichia coli* [35]. The results obtained by the scientists show that preparations obtained from raspberry leaves can complement some classical antibiotics.

Also, the scientific literature analysis revealed that antibacterial activity values of raspberry plant parts show great differences and fluctuations. It is clear that even with the same analytical methods, the values obtained are affected by a large variability due to many factors, such as the part of the plant used (fruits, leaves, and seeds), the principles and environment of raspberry cultivation, meteorological and climatic conditions, sample preparation methodology, and the solvent used. Most of the researchers agree that raspberry plant compounds, as natural product-based compounds, present anti-toxin properties. Even though these scientific reports have highlighted the antimicrobial activity of raspberries, studies to also evaluate the potential therapeutic use and application in other industries of all plant parts would be necessary.

Our tests of antimicrobial activity are presented in Table 1 in terms of the inhibition zone diameter (mm) of extracts from test samples against Gram-positive (*S. aureus*, *B. subtilis*, *E. faecalis*, and *L. monocytogones*) and Gram-negative (*S. Typhimurium*, *P. aeruginosa*, *E. aerogenes*, and *E. coli*) bacterial strains using raspberry inflorescence, puree, unripe fruit, leaf, stem, and root extracts.

In the present study, all the extracts isolated from raspberries fruits, inflorescences, puree, unripe fruits, leaves, seeds, stems, and roots possessed antimicrobial activity against all the tested bacteria except *E. aerogenes* and *E. coli*, which were the most resistant to the applied raspberry plant extracts. Only raspberry puree showed activity against *E. aerogenes*, and raspberry puree, seeds, and roots exhibited activity against *E. coli*. 

One of the most sensitive bacteria to raspberry plant parts was determined to be *S. aureus*, followed by *L. monocytogenes* and *B. subtilis*. This study discovered that Gram-positive bacteria are more sensitive to raspberry plant parts extracts. The highest inhibitory effect against Gram-positive bacteria *Staphylococcus aureus* was that of the extracts isolated from raspberry inflorescence and unripe berries. 

Extracts from different raspberry plant parts showed excellent antibacterial activity against Gram-positive bacteria (with zone of inhibition ranging from 9.0 to 21.0 mm), but they did not exhibit significant antibacterial activity against Gram-negative bacteria, especially *E. aerogenes* and *E. coli*. Similarly, De Santis et al. [34] did not observe any inhibitory effect of raspberry leaf extract against *E. coli* as determined by the well diffusion method. Our research revealed that only raspberry puree exhibited inhibitory effects against *E. aerogenes*, while ripe fruits, seeds, and roots could inhibited *E. coli*.

Krstić et al. proved that the results of research on antibacterial activity highly dependent on the method and sample preparation [10], so it would be inappropriate to compare the results of our study with those of other authors. Therefore, in the future, research on the development and modelling of production processes for products with antimicrobial properties would be very important. The differences in antibacterial potential shown by different authors may also be due to the procedure of extraction, type of solvent used, geographical origin of plants, or time of leaf harvesting. As can be seen in our research, the tested Gram-negative bacteria *Staphylococcus aureus* and *L. monocytogones* were the most resistant to the applied raspberry plant parts extracts. The raspberry parts extracts, with the exception of the puree, were inactive against *E. aerogenes*.

### 2.2. Total Phenolic Content and Antioxidant Activity Correlation with Antimicrobial Activity

Qualitative and quantitative analyses of phenolic compounds in raspberry plant parts were carried out, and raspberry extracts were characterized in our previous studies [35]. For the purpose of better understanding the antibacterial properties and their correlation with total phenolic content and antioxidant activity, we investigated the percentage distribution of phenolic compounds in raspberry plant parts in manufacturing processes. 

Since raspberry by-products are generated during primary production (cultivation) and secondary processing, in this part, we separated raspberry plant parts by classifying the samples as follows: by-products from primary production (where the main product is fresh raspberries) are leaves, stems, flowers, roots, and buds, and by-products from secondary production (where the main product is juice) are seeds. In this study, we examined the percentage distribution of phenolic compounds in manufacturing processes. The results of the high-performance liquid chromatography–photodiode array detection (HPLC-PDA) method of analysis showed that seventeen phenolic compounds, belonging to subgroups of simple phenols, flavonols, and proanthocyanidins, were detected in raspberry plant parts (Figure 1 and Figure 2).

Significantly, the highest contents of tiliroside and kaempferol-3-O-glucuronide were determined in raspberry leaves, whereas epicatechin and procyanidin C1 were found in raspberry roots.

Salicylic acid was an abundant phenolic compound in berries (without seeds) and seeds (Figure 2). Isoquercetin was found in 100% of berries (without seeds) but not in the seeds. These results demonstrate the importance of separating plant parts to obtain higher levels of a specific phenolic compound in the final product. The individual phenol content is completely different between different raspberry plant parts. These results suggest that raspberry plant parts could be used as a nutraceutical resource and a functional food ingredient. These bioactive compounds found in different plant parts exert their beneficial biological effects and hence may promote human health through different mechanisms of action.

In this paper, we also aimed to expand research knowledge by defining the correlation between TPC, DPPH, ABTS, and FRAP values and antimicrobial activity. Total phenolic content and antioxidant activity varied by plant part, and the correlation with antimicrobial activity depended on the tested bacterium. No significant correlation was found with *E. aerogenes* and *E. coli* bacteria. They can grow under aerobic and anaerobic conditions and do not produce enterotoxins. The results are presented in Figure 3 below.

According to Pearson correlation interpretation, the correlation coefficient ranges from −1 to 1, and the absolute value of exactly 1 implies that a linear equation describes the relationship between X and Y perfectly, with all data points lying on a line; in light of this, it can be concluded that the total amount of phenolic compounds or antioxidant activity of plant parts did not show the highest correlation with antibacterial activity.

The correlation coefficients of total phenolic content (TPC) and the antimicrobial activity of plant parts were calculated for every bacterial strain tested. The value arrays used were TPC content of different plant parts (berries, stems, inflorescences, roots, leaves, and seeds) to the antimicrobial activity of each plant part. The data in Figure 3 show a moderate positive correlation between TPC value and antimicrobial activity against *S. aureus* (0.487) and *B. subtilis* (0.475); a low correlation for *E. faecalis* (0.190), *L. monocytogenes* (0.222), *S. typhimurium* (0.303), and *P. aeruginosa* (0.278); and no correlation for *E. aerogenes* and *E. coli*. The higher the antioxidant level, the better the antibacterial activity obtained from the corresponding extracts. The concentration of phenolic compounds and antioxidant activity in a sample has been shown to be highly correlated with its antibacterial potential. The higher the phenolic contents and antioxidant activity, the higher the antibacterial potential. 

We determined the correlation coefficients for antioxidant activity (using DPPH, ABTS, and FRAP assays) and the antimicrobial activity of plant parts for every bacterial strain tested (the value arrays used were DPPH assay values of different plant parts (berries, stems, inflorescences, roots, leaves, and seeds) to the antimicrobial activity of each plant part). We found a moderate positive correlation between the FRAP assay value and antimicrobial activity against *S. aureus* (0.573) and *B. subtilis* (0.565); a low positive correlation between the DPPH assay value and antimicrobial activity against *S. aureus* (0.353) and *B. subtilis* (0.275); and negligible to no correlation for *E. faecalis*, *L. monocytogenes*, *S. typhimurium*, *P. aeruginosa*, *E. aerogenes*, and *E. coli* for all assays. 

### 2.3. Raspberry Plant Parts’ Potential for Added-Value Products

The study results suggest raspberries may not be effective solution to combat Gram-negative bacteria, which are known to be highly resistant to various antibiotics and natural remedies from plants. All parts of the raspberry plant would be effective in the development of drugs against *S. aureus*, *L. monocytogenes*, *S. typhimurium*, and *B. subtilis* bacteria. Raspberry leaves, blossoms, stems, and even roots can be used prophylactically or alongside treatment for diseases such as pneumonia and bloodstream infections. Also, in the future, it would be useful to conduct research and develop disinfectants that could be used at home or even in hospitals in the fight against *E. faecalis* bacteria. Raspberry farms can use extracts for the disinfection of equipment, packaging, or containers according to the principles of the circular economy.

The findings of this research also indicate that extracts from different raspberry plant parts have tremendous potential as a natural product with antibacterial capabilities and can be used to maintain food or drink quality, as animal feed, in cosmetics, in other industries (e.g., manufacture of disinfectants), and for packaging. As expected, extracts from different plant parts of raspberry exhibited different antibacterial effects. It is evident which parts of the plant are the most effective against the tested bacteria, guiding manufacturers in producing products that meet the required quality indicators. The results obtained from studies on the antibacterial potential of raspberry plant parts as added-value products reveal the following findings. 

The unripe fruit extracts showed the highest antimicrobial activity against *B. subtilis, E. faecalis*, and *S. typhimurium*. Those bacteria have been assigned a high and even critical (*P. aeruginosa*) level on the global priority pathogens list of antibiotic-resistant bacteria by the WHO. This indicates that unripe raspberries have the potential for the development of new and effective products for disease prevention or antibiotic treatments. However, ripe raspberries or raspberry puree is the best and must always be used as a preventative measure during periods when people are the most susceptible to infections and as a diaphoretic agent. Raspberries were the only parts of the plant that defeated *E. aerogenes* and *E. coli* bacteria and also showed the best activity against anaerobic bacterium *L. monocytogenes*.

It should also be noted that raspberry inflorescences and leaves have a good antibacterial effect, and since these parts are considered by-products in primary production, they should be used to create a product in order to develop waste-free technologies on farms. The inflorescence of raspberries is the best against *S. aureus* and very effective against *B. subtilis, L. monocytogenes*, and *S. typhimurium*. In contrast, the extracts from roots and seeds had the lowest antibacterial activity compared with other plant parts. Therefore, when optimizing processing and product production, these parts of the plant should be used for their other good properties. 

Given the importance of the “farm-to-fork” approach in the control of food-borne diseases, the development of raspberries plant part-synthesized extracts and their employment into different industries can be effective strategies against *S. aureus*, *B. subtilis*, *L. monocytogones*, *P. aeruginosa*, and *S. typhimurium* bacteria, which are responsible for human illnesses, thus addressing public health concerns about infection. By using concentrated extracts from different raspberry plant parts, it is possible to create new products by developing a circular economy. The production of raw materials with highly effective antimicrobial properties from plants is very important in the current trend of sustainable development. Products created from raspberry plant parts can serve as a potential antidote against bacterial infections. 

According to research results, raspberry plant parts can be used in food or cosmetics products and as health-promoting agents to treat diseases, and functional extracts can be incorporated into films and used in packaging. The added-value products from parts of the raspberry plant can also be created in other industries. Such products would have the greatest demand among consumers who prefer sustainable, ecological products. In the future, it will be necessary to determine effective application dosages and optimize production processes to ensure economic viability. Additionally, conducting comparative studies with other plant materials would be valuable.

It is worth mentioning that there is still a lack of application of such natural antimicrobial production, and the establishment of such products in the market requires scientists and practitioners to solve questions related to aspects such as quality, e.g., sensory properties; the activity and stability of antimicrobials during processing; and the standardization of methods for processing. It is important to determine the information given to the end user in the form of label data to make economic calculations. It is also recommended to carry out scientific research in terms of toxicity studies and to determine the safety of such economically beneficial and processing technologies. The conducted study significantly complements the work of other scientists by separately reporting on the antibacterial activity of different parts of raspberry plants grown under the same conditions and shows the biological activity potential characteristics of all raspberry plant parts. This allows for further research on the development of new products for the food, cosmetic, and other industries.

## 3. Materials and Methods

### 3.1. Chemicals and Reagents

We used solvents and reagents, including ethanol (96%; Stumbras, Kaunas, Lithuania). Methanol (99%), anhydrous acetic acid (99.8%), and hydrochloric acid (37%) were purchased from Sigma–Aldrich (Buchs, Switzerland). The following reagents were used: 2,2′-azino-bis(3-ethylbenzothiazoline-6-sulfonic acid) diammonium salt (ABTS), 2,4,6-Tri-(2-pyridyl)-S-triazine (TPTZ), 2,2-Diphenyl-1-picrylhydrazyl (DPPH)ferric chloride hexahydrate (FeCl_3_ × 6H_2_O), sodium acetate (CH3COONa), Folin–Ciocalteu’s phenol reagent, potassium persulfate (K_2_S_2_O_8_), anhydrous ferrous chloride (FeCl_2_), and 6-hydroxy-2,5,7,8-tetramethylchroman-2-carboxylic acid (Trolox) from Sigma–Aldrich (Buchs, Switzerland).

Gram-positive *Staphylococcus aureus* (ATCC 25,923), *Bacillus subtilis* (ATCC 6633), *Enterococcus faecalis* (ATCC 19,433), and *Listeria monocytogenes* (ATCC 13,932) and Gram-negative *Salmonella typhimurium* (ATCC 14,028), Pseudomonas aeruginosa (ATCC 27,853), *Enterobacter aerogenes* (ATCC 13,048), and *Escherichia coli* (ATCC 8739) bacterial test cultures were used. In addition, two strains of yeast were used, *Candida albicans* (ATCC 10,231) and *Saccharomyces cerevisiae* (ATCC 9763), obtained from UAB “Diamedika”, Vilnius, Lithuania. Potato glucose dextrose agar slants and potato dextrose agar were obtained from Liofilchem, Roseto degli Abruzzi, Italy, and 90 mm diameter Petri dishes, from APTACA, Canelli, Italy. Bacterial cultures were grown for 18 h at 37 °C on agar slants (plate count agar; Liofilchem, Italy).

### 3.2. Material and Its Preparation

For the analyses, raspberries were obtained from a berry farm located in Šiauliai district, Lithuania. Fruits, inflorescences, leaves, stems, and roots of raspberry cultivar “*Polka*” were collected during harvest, in August 2022. This variety is an early-cropping primocane; its fruits have firm flesh and can be stored very well, for a raspberry. Raspberries were collected from 5 randomly selected locations in an area of 1 hectare. From the total sample, the amount of 400 g was taken for chemical analyses. Chemical analyses of raspberry morphological parts were performed in three repetitions. Morphological plant parts were divided from the plant manually. Raspberry seeds were separated from the pulp through a sieve. The raw material was frozen, lyophilized, and ground.

Ten grams of freeze-dried raspberries of different dry and ground morphological parts was mixed with 50 mL of 70% ethanol. These samples were left for 24 h in the dark. Afterwards, the samples were centrifuged for 10 min at 8500 rpm in a Biofuge Stratos centrifuge and filtered. The ethanol from extracts was evaporated in a rotary vacuum evaporator, Büchi R-250 (Büchi Laboratortechnic, Flawil, Switzerland). 

### 3.3. Microorganisms and Preparation of Extracts for Antibacterial Testing

Bacterial cultures were obtained from the Food Institute of Kaunas University of Technology (Lithuania). The strains were maintained on agar slants at 4 °C and activated at 37 °C on nutrient agar for 24 h before any susceptibility test. Dry extracts, made of lyophilized and ground raspberry plant parts, were re-dissolved in 80% methanol to produce 10% solutions. 

The assessment of the antimicrobial activity of the prepared extracts was performed on four Gram-positive (*Staphylococcus aureus*, *B. subtilis*, *E. faecalis*, and *L. monocytogenes*) and four Gram-negative (*S. typhimurium*, *P. aeruginosa*, *E. aerogenes*, and *E. coli*) bacterial test cultures. In addition, two strains of yeast were used: *C. albicans* and *S. cerevisiae*.

Bacterial cultures were grown for 18 h at 37 °C on slant agar surfaces for the assessment of antibacterial activity through agar diffusion. The washed bacterial suspension was diluted according and mixed well with a mini shaker, and the appropriate number of cells was added to an agar medium. The mixture of bacterial cell suspension prepared in this way with the medium was poured into 90 mm diameter glass Petri dishes in 10 mL increments. Yeast cultures were grown at 25 °C for one day on a potato glucose agar slant. After one day, grown yeast cultures were washed from the agar using a sterile physiological solution and according to McFarland standard No. 1. The prepared cell suspension was poured into the dissolved medium of potato glucose agar, cooled, and mixed well. The mixture of bacterial and yeast cell suspension prepared in this way with the medium was poured into 90 mm diameter Petri dishes in 10 mL increments. After the medium solidified, 6 wells (8 mm in diameter) were filled with it.

The antimicrobial effect on bacterial cultures was evaluated after 6 days of incubation, according to the diameter of the clear zones formed around the wells, expressed in millimetres. If clear zones failed to form around the wells, it was concluded that the tested substance or its concentration did not have a bactericidal effect on the tested culture.

### 3.4. Antioxidant Activity (DPPH, ABTS, and FRAP), Total Phenolic Content (TPC), and HPLC Analysis

Antioxidant activity and total phenolic content (TPC) were determined using the methodologies described in Appendix A.

### 3.5. Statistical Analysis

All the experiments were carried out in triplicate. The standard deviation was calculated and presented together with the mean values. Pearson correlation analysis between total phenolic content and bioactivities was carried out at a 95% confidence level. IBM SPSS Statistics 26 (USA) and Excel 2021 (USA) software packages were used for statistical analysis. Differences were considered to be significant at *p* < 0.05.

## 4. Conclusions

In this study, the antibacterial activity of different morphological parts of raspberries revealed their potential as added-value products. The extracts demonstrated excellent antibacterial efficacy against Gram-positive bacteria, with values ranging from 9.0 to 21.0 (zone of inhibition in mm). However, they did not exhibit significant antibacterial inhibition area diameter (mm) against Gram-negative bacteria, except for *S. typhimurium*, the activity against which ranged from 9.0 to 18.3 (zone of inhibition in mm). Notably, *E. aerogenes* and *E. coli* were particularly resistant. Raspberry plant parts did not show biological activity against *C. albicans*, *S. cerevisiae*, *E. coli*, or *E. aerogenes*, except for raspberry puree. Among the various plant parts, root and seed extracts had the lowest antibacterial activity. This study’s results indicated that raspberry extracts from different parts could effectively combat three Gram-positive bacteria. Data revealed a moderate positive correlation between total phenolic content (TPC) and antimicrobial activity against *S. aureus* (0.487) and *B. subtilis* (0.475); a low correlation with *E. faecalis* (0.190), *L. monocytogenes* (0.222), *S. typhimurium* (0.303), and *P. aeruginosa* (0.278); and no correlation for *E. aerogenes* and *E. coli*. 

The determined correlation coefficients of antioxidant activity and the antimicrobial activity of plant parts show a moderate positive correlation between FRAP assay value and antimicrobial activity against *S. aureus* (0.573) and *B. subtilis* (0.565); a low positive correlation between DPPH assay value and antimicrobial activity against *S. aureus* (0.353) and *B. subtilis* (0.275); and negligible to no correlation for *E. faecalis*, *L. monocytogenes*, *S. typhimurium*, *P. aeruginosa*, *E. aerogenes*, and *E. coli* for all assays. 

These findings underscore the viability of substituting synthetic antimicrobials with natural alternatives. Raspberry plant parts can find applications in emerging fields that generate economic, environmental, and social value. It can be concluded that the different raspberry plant parts examined, which are usually thrown away in the fields after secondary processing, can be utilized as highly valuable and inexpensive raw materials for the production of functional food products or pharmaceuticals. These findings can form the basis for further studies to isolate active compounds and evaluate them against bacteria with the goal of finding new antimicrobial medicines.

## Figures and Tables

**Figure 1 plants-13-00504-f001:**
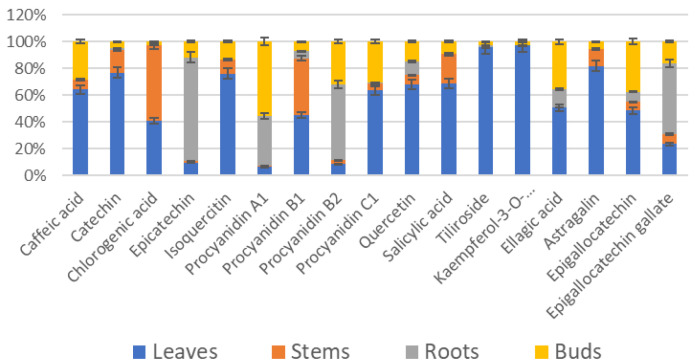
Phenolic compounds (%) in raspberry by-products from primary production.

**Figure 2 plants-13-00504-f002:**
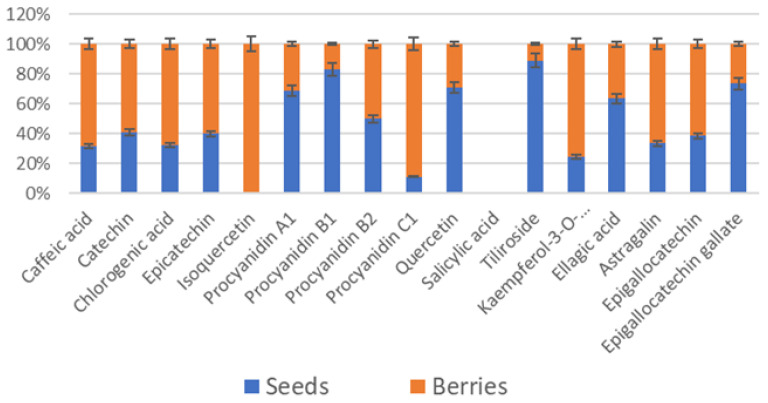
Phenolic compounds (%) in raspberry by-products from secondary production.

**Figure 3 plants-13-00504-f003:**
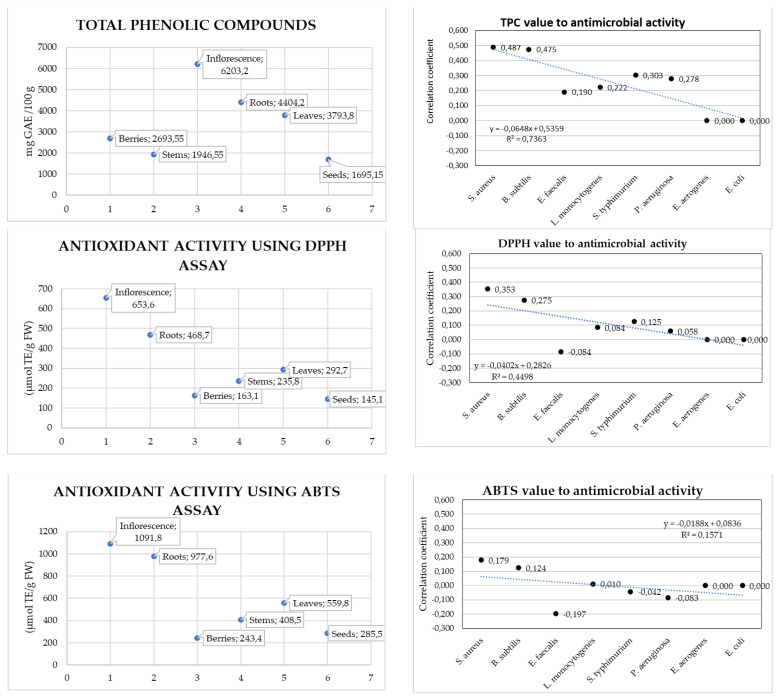
The total phenolic content; antioxidant activity using DPPH^•^, ABTS^•+^, and FRAP assays; and correlation coefficients of the antimicrobial activity of raspberry plant parts for every bacterial strain tested.

**Table 1 plants-13-00504-t001:** Antibacterial activity of various extracts of test samples against bacterial species.

	Sample	Inflorescences	Raspberry Puree	Unripe Raspberries	Leaves	Seeds	Stems	Roots
*Gram-positive bacteria*	*S. aureus*	21.0 ± 1.1	18.0 ± 0.9	20.3 ± 1.0	17.3 ± 0.9	13.6 ± 0.7	17.0 ± 0.9	14.6 ± 0.7
	*B. subtilis*	16.0 ± 0.8	15.3 ± 0.8	17.0 ± 0.9	12.3 ± 0.6	9.0 ± 0.4	10.0 ± 0.5	10.0 ± 0.5
	*Enterococcus faecalis*	12.3 ± 0.6	15.3 ± 0.8	16.3 ± 0.8	13.0 ± 0.7	9.0 ± 0.4	9.0 ± 0.4	9.0 ± 0.4
	*L. monocytogenes*	18.0 ± 0.9	21.0 ± 1.1	17.3 ± 0.9	20.0 ± 1.0	16.0 ± 0.8	18.0 ± 0.9	16.0 ± 0.8
*Gram-negative bacteria*	*Salmonella typhimurium*	17.3 ± 0.9	18.0 ± 0.9	18.3 ± 0.9	16.0 ± 0.8	9.0 ± 0.4	14.0 ± 0.7	9.0 ± 0.4
	*P. aeruginosa*	12.6 ± 0.6	12.3 ± 0.6	15.0 ± 0.8	10.6 ± 0.5	9.0 ± 0.4	9.0 ± 0.4	9.0 ± 0.4
	*E. aerogenes*	0.0	9.0 ± 0.4	0.0	0.0	0.0	0.0	0.0
	*E. coli*	0.0	14.0 ± 0.7	0.0	0.0	9.0	0.0	9.0
	*Candida albicans*	0.0	0.0	0.0	0.0	0.0	0.0	0.0
	*Saccharomyces cerevisiae*	0.0	0.0	0.0	0.0	0.0	0.0	0.0

*Note*: The table presents data on the measured inhibition zones, in mm. The data are expressed as average values and standard deviations of three replicates and indicate significant differences (*p* < 0.05).

## Data Availability

The data presented in the present study are available in the article.

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
