# Peer review of "Correlation between Antimicrobial Activity Values and Total Phenolic Content/Antioxidant Activity in Rubus idaeus L."

_plants, 2024, doi:10.3390/plants13040504_

Round 1
Reviewer 1 Report
Comments and Suggestions for Authors
Please explain the "Pearson correlation explanation" in detail and provide a detailed description of the relationships in the correlation chart, including the unlabelled extracts from different parts of the raspberry in the correlation chart.
Please label the order of the charts.
Comments on the Quality of English Languageno comment
Author Response
Thank You for your beneficial comments.
The all suggestions were included in the article. Also:
- We’ve revised and made some changes in the introduction and highlighted the importance of the study.
- We’ve revised the references and did some changes in all the paper
- We’ve improved methods description
- We’ve made some changes in the results and conclusions sections, provided some additional information
Reviewer 2 Report
Comments and Suggestions for Authors
The manuscript “Antibacterial properties of different morphological parts of Rubus idaeus L.” describes the antibacterial potency of raspberry leaves, inflorescence, and fruits. The most promising results were achieved toward three Gram-positive bacteria. So, raspberry plant parts can find applications in emerging fields that generate economic and environmental value. The manuscript is well and carefully written, data are solid and conclusions are justified by the results. At the same time some corrections should be done. It seems that Figure 4 is unnecessary because it is loosely coupled with the obtained results and their discussion. And the final, why the antimicrobial screening is resulted in measuring of “antibacterial diameter” but not IC50 values?
Author Response
First of all, the authors sincerely thank you for such an excellent evaluation of the article. We’ve followed your suggestion and delated figure 4.
Please, find below our answer to your question:
In this phase of the research, the search for extracts with antimicrobial effects was carried out in order to find the morphological parts of raspberries with antimicrobial effects. Other research methods will be used in further research and comparative studies.
Reviewer 3 Report
Comments and Suggestions for Authors
Manuscript ID
plants-2866114
Antibacterial properties of different morphological parts of Rubus idaeus L.
The study discusses the antibacterial activity of different plant parts of Rubus idaeus L., its correlation with total phenolics, and its potential for value addition.
Studies by Schulz and Chim 2019, Krauze-Baranowska and coworkers 2019, discussed the biological activities of the Rubus genus, what was the rationale for focusing on the antibacterial activity of the plant? Discuss.
Line 19, New aspects were revealed for further use and making novel and natural products. How is the present study significant in preparing novel products as antibacterials? Discuss the prospects for future utilization.
Line 23-24, These findings underscore 23 the viability of substituting synthetic antimicrobials with natural alternatives.
While both natural and synthetic antimicrobials have been potent in tacking drug-resistant microbes, natural products and their derivatives, show potent efficacy as antimicrobials. List some examples of plant-based antimicrobials marketed as commercial antimicrobial drugs.
Line 97-99, One approach to 97 creating functional food is to utilize by-products from the agro-food sector that are considered waste but are rich in bioactive compounds.
This is a different area of utilizing plant-by-products for value addition and is not relevant in the present context. Please delete.
What standard was used to compare and determine the antimicrobial activity of the plant parts?
The manuscript needs to be reorganized and rewritten for clarity and improved language.
Line 16-17- Plant by-products, which are thrown into the environment, are rich in valuable com- 16 pounds???? The sentence does not make sense, and is not relevant in the context of the manuscript. Revise.
English language needs to be thoroughly improved for consideration, there are many grammatical mistakes, unclear sentences etc.
Comments on the Quality of English Language
Extensive English revision is required.
Author Response
Thank You for the constructive and beneficial comments. The all your suggestions were included in the article. Please, find our answers to your comments in the table below.
|
1. Studies by Schulz and Chim 2019, Krauze-Baranowska and coworkers 2019, discussed the biological activities of the Rubus genus, what was the rationale for focusing on the antibacterial activity of the plant? Discuss. |
We’ve discussed as you suggested. |
|
2. Line 19, New aspects were revealed for further use and making novel and natural products. How is the present study significant in preparing novel products as antibacterials? Discuss the prospects for future utilization. |
We did as you suggested |
|
3. Line 23-24, These findings underscore 23 the viability of substituting synthetic antimicrobials with natural alternatives. |
We’ve corrected |
|
While both natural and synthetic antimicrobials have been potent in tacking drug-resistant microbes, natural products and their derivatives, show potent efficacy as antimicrobials. List some examples of plant-based antimicrobials marketed as commercial antimicrobial drugs. |
We did as you suggested |
|
Line 97-99, One approach to 97 creating functional food is to utilize by-products from the agro-food sector that are considered waste but are rich in bioactive compounds. This is a different area of utilizing plant-by-products for value addition and is not relevant in the present context. Please delete. |
We did as you suggested |
|
What standard was used to compare and determine the antimicrobial activity of the plant parts? |
No standard was used to evaluate the antimicrobial activity. The results were compared with each other in terms of the antimicrobial effectiveness of the extracts of individual parts of the plant, in order to find the most effective extracts, while also determining the sensitivity characteristics of individual microorganisms to the extracts. |
|
The manuscript needs to be reorganized and rewritten for clarity and improved language. English language needs to be thoroughly improved for consideration, there are many grammatical mistakes, unclear sentences etc. |
The revision in all aspects was done in the article, with the consultation of native English speakers and all co-authors. |
|
Line 16-17- Plant by-products, which are thrown into the environment, are rich in valuable com- 16 pounds???? The sentence does not make sense, and is not relevant in the context of the manuscript. Revise. |
We’ve corrected it. |
Reviewer 4 Report
Comments and Suggestions for Authors
I think this research is significant enough but there is a sufficient room for improvement if the following omissions are corrected:
The title of the article is not descriptive enoung. The authors did not mention the antioxidant activity of R. idaeus and its correlation to antibacterial activity which are also determined in this study.
In Introduction the authors did not describe the plant tested, its family and pharmacological activities, one of which is the antimicrobial activity. I think the information between lines 68-95 belongs to the Discussion and in Introduction the authors could summarize it in several sentences, if they found it necessary.
Line 90 - R. idaeus should be in Italic. Please check the article for such omissions.
Lines 120-125 - the binominal names of bacteria and yeasts should be in Italic. Moreover, this data is repetitive (Lines 150-156).
Subsection 3.1 (Line 184) contains results which, in my opinion, are insufficiently discussed. For example, are the zones of inhibition similar to these of other authors or there are differences? And what is the reason for such differences? Subsection 3.2 data could be more thoroughly discussed as well. Such discussion can help the readers to place the data on the proper place among similar investigations.
Lines 196-200 and 231-233 contain similar information. I propose this information to be summarize in one place, better to the end of subsection where lines 231-233 are.
Lines 307-310 - The content of Figure 4 is not very clear. Moreover, I think the information about these 6 bacteria is not necessary to be included in this paper.
Author Response
Thank You for such good, constructive and beneficial comments. The all your suggestions were included in the article. Please, find our answers to your comments in the table below.
|
The title of the article is not descriptive enoung. The authors did not mention the antioxidant activity of R. idaeus and its correlation to antibacterial activity which are also determined in this study. |
We’ve changed the title of the article |
|
In Introduction the authors did not describe the plant tested, its family and pharmacological activities, one of which is the antimicrobial activity. I think the information between lines 68-95 belongs to the Discussion and in Introduction the authors could summarize it in several sentences, if they found it necessary. |
We’ve corrected as you suggested |
|
Line 90 - R. idaeus should be in Italic. Please check the article for such omissions. |
We did it |
|
Lines 120-125 - the binominal names of bacteria and yeasts should be in Italic. Moreover, this data is repetitive (Lines 150-156). |
We wrote names in Italic and corrected the repetitive data |
|
Subsection 3.1 (Line 184) contains results which, in my opinion, are insufficiently discussed. For example, are the zones of inhibition similar to these of other authors or there are differences? And what is the reason for such differences? Subsection 3.2 data could be more thoroughly discussed as well. Such discussion can help the readers to place the data on the proper place among similar investigations. |
We’ve corrected as you suggested |
|
Lines 196-200 and 231-233 contain similar information. I propose this information to be summarize in one place, better to the end of subsection where lines 231-233 are. |
We did as you suggested |
|
Lines 307-310 - The content of Figure 4 is not very clear. Moreover, I think the information about these 6 bacteria is not necessary to be included in this paper. |
We’ve deleted the Figure 4 |
Round 2
Reviewer 1 Report
Comments and Suggestions for Authors
The content that has been appropriately refined and summarized.
Comments on the Quality of English Languagenone
Author Response
Thank you for your comments. The authors have taken your comments into account and carefully reviewed the entire text again.
Reviewer 3 Report
Comments and Suggestions for Authors
Thank you for revising the manuscript.
Author Response

(The authors gave the same response as above.)
